# Non-Stationary Stochastic Global Optimization Algorithms

Jonatan Gomez * and Andres Rivera

Departamento de Ingeniería de Sistemas e Industrial, Facultad de Ingeniería, Universidad Nacional de Colombia, Bogotá 11001, Colombia
* Correspondence: jgomezpe@unal.edu.co

**Abstract:** Studying the theoretical properties of optimization algorithms such as genetic algorithms and evolutionary strategies allows us to determine when they are suitable for solving a particular type of optimization problem. Such a study consists of three main steps. The first step is considering such algorithms as Stochastic Global Optimization Algorithms (SGOALs), i.e., iterative algorithm that applies stochastic operations to a set of candidate solutions. The second step is to define a formal characterization of the iterative process in terms of measure theory and define some of such stochastic operations as stationary Markov kernels (defined in terms of transition probabilities that do not change over time). The third step is to characterize non-stationary SGOALs, i.e., SGOALshaving stochastic operations with transition probabilities that may change over time. In this paper, we develop the third step of this study. First, we generalize the sufficient conditions convergence from stationary to non-stationary Markov processes. Second, we introduce the necessary theory to define kernels for arithmetic operations between measurable functions. Third, we develop Markov kernels for some selection and recombination schemes. Finally, we formalize the simulated annealing algorithm and evolutionary strategies using the systematic formal approach.

**Keywords:** evolutionary algorithms; non-stationary markov kernel; convergence analysis; evolutionary strategies; simulated annealing; selection schemes, recombination schemes, stochastic optimization



## 1. Introduction

In global optimization studies, a general question is which type of algorithms are more fit for which type of optimization problems, so, we can decide when to use an evolutionary algorithm to solve a class of optimization problems by understanding its theoretical properties and characterizing its observable behavior [1].

According to Zilinskas and Zhigljavsky in [2], stochastic global optimization algorithms (SGOALs in short) are inseparable from their presentation and analysis. Several researchers have done active development of the field from long time ago. Such as: Torn and Zilinskas in [3], Mockus and Zilinskas in [4], or Neimark and Strongin in [5]. However, it remains an active field of research including mathematical analysis of problems.

SGOALshave been also studied from a Markovian perspective: Zhigljavsky and Zilinskas in Sections 3.3 and 3.4 in [6] and Tikhomirov in [7] studied the convergence rate of some homogeneous Markov monotone random search optimization algorithms. Also, H Al-Mharmah et al. in [8] studied some random non-adaptive algorithms for finding the maximum of a continuous function on the unit interval. An analysis of selection algorithms was done by Chakraborty et al. in [9] and an analysis for evolutionary strategies for global minimization was described by François in [10].

As can be noticed, these studies do not use measure theory to formalize probabilistic concepts or are developed around a specific optimization problem. Gomez in [11], describes a formal and systematic approach for characterizing stochastic global optimization algorithms. There, the required theory of probability to characterize SGOALssuch as measure theory, Markov kernel, operations between kernels, products and conditions to study

convergence is presented. In addition, it is proved that some algorithmic functions like projection and sort can be represented by kernels.

Moreover, the notion of join-kernel has been introduced in that paper as a way to characterize the combination of stochastic methods. Also, it is defined a formal structure of an optimization space for studying SGOALs. Finally, Gomez formalizes algorithms that have a next-population stochastic method, which does not change transition probabilities over time. Such algorithms can be viewed from the perspective of stationary Markov processes. This viewpoint applies among others for, standard versions of hill-climbing, parallel hill-climbing, steady-state genetic, generational genetic, and differential evolution algorithms.

This work continues such a systematic formal approach. First, we review the theory done by Gomez in [11]. Next, we generalize the sufficient conditions convergence Lemma 71 in [11] from stationary to non-stationary Markov processes. Third, we develop arithmetic kernels to characterize arithmetic operations between measurable functions. We develop Markov kernels for some selection and recombination schemes. Finally, in order to show some applications of the concepts developed, we formalize both simulated-annealing and evolutionary-strategies using the systematic formal approach , which are classical algorithms and can be found several studies in the literature such as Romeijn et al. in [12] or Weise in [13] .

## 2. Preliminaries

This section provides a brief introduction to the systematic formalization proposed by Gomez in [11]. Such systematic formalization of SGOALs, is carried on Markov kernels terms. We formalize SGOALs with stationary next population stochastic method, i.e., SGOALs that can be characterized as stationary Markov processes and do not change transition probabilities of the next population over time. That is the case of the hill-climbing [14], the parallel hill-climbing, the generational genetic [15–17], the steady-state genetic [18], and the differential evolution [19,20] algorithms. However, SGOALs such as the Simulated Annealing [21], Evolutionary Strategies [22], or any algorithm using parameter control/adaptation techniques [23] cannot be characterized as stationary Markov processes.

We clarify that in this section we only review the concepts and not the proofs. The proof of each concept can be found in [11].

### 2.1. Systematic Formalization Theory

We review the concepts used to characterize SGOALsand necessary concepts to extend the theory to characterize adaptive SGOALs.

We consider an optimization problem with an objective function $f : \Phi \to \mathbb{R}$, which is defined over a feasible region $\Omega \subseteq \Phi$, where $\Phi$ is the solution space and $f$ is a function that is looking for a global optimizer $p^*$ described by:

$$\min(f : \Phi \to \mathbb{R}) = \{p^* \in \Omega \subseteq \Phi \mid (\forall p \in \Omega)(f(p^*) \leq f(p))\}. \tag{1}$$

#### 2.1.1. Stochastic Global Optimization Algorithm

In this work, we focus on algorithms that are not deterministic but stochastic. e.g., simulated annealing and evolutionary strategies. A generic way to describe such algorithms in pseudocode is given by Algorithm 1. Here the main difference between algorithms is the way the NEXTPOP operation is carried out.

The NEXTPOP method generates new populations, the INITPOPN($n$) method generates an initial population and BEST($P_t$) chooses the best set of individuals from $P_t$.

#### 2.1.2. Measure and Probability Theory

Gomez in [11] used probabilistic kernels to formalize SGOALs. This work is an extension of Gomez's work. We review some concepts that Gomez uses and that we are going to use to characterize adaptive SGOALs.

---

**Algorithm 1** Stochastic Global Optimization Algorithm

---
SGOAL(*n*)
1. $t_0 = 0$
2. $P = \text{INITPOP}(n)$
3. **while** $\neg\text{END}(P_t, t)$ **do**
4.    $P_{t+1} = \text{NEXTPOP}(P_t)$
5.    $t = t + 1$
6. **return** $\text{BEST}(P_t)$

---

Definitions of Measure Theory

Probability theory uses measure theory to formalize the concepts. Measure theory defines elementary events $\Omega \neq \varnothing$ and the system of observable events $\mathcal{A} \subseteq 2^{\Omega}$, where $\mathcal{A}$ is a family of sets. These concepts can be translated in the context of SGOAL where the set of elementary events contains all possible populations and, a family of sets of populations is a system of observable events.

Probability theory operates over sets to measure the probability of some observable event. The structure used to measure subsets of $\Omega$ is a $\sigma$-algebra that meets the following conditions:

1. $\Omega \in \Sigma$. $\Omega$ is considered a universal set.
2. $\Sigma$ is $\overline{C}$. $\Sigma$ is closed under complement.
3. $\Sigma$ is $\overline{CU}$. $\Sigma$ is closed under countable unions.

When we deal with a continuous space, a topological space is used . Hence, $(\Omega, \tau)$ is a topological space, the sigma-algebra ($\sigma$-algebra) $\mathcal{B}(\Omega) \equiv \mathcal{B}(\Omega, \tau) \equiv \sigma(\tau)$ is the Borel $\sigma$-algebra on $\Omega$. Consider set $\Omega = \mathbb{R}$, in that case $\mathcal{B}(\mathbb{R})$ is the $\sigma$-algebra on $\mathbb{R}$. In this paper, a tuple of the form $(\Omega, \Sigma)$, refers to a measurable space, where $\Sigma$ is a $\sigma$-algebra.

In probability theory, we generally need to operate between measurable spaces using certain functions, precisely measurable functions. Let $(\Omega_1, \Sigma_1)$ and $(\Omega_2, \Sigma_2)$ be two measurable spaces. A function $f$ is defined as measurable if $f: \Omega_1 \to \Omega_2$, and $f^{-1}(B)$ maps a subset of the domain. i.e., if $B \in \Sigma_2$ then $f^{-1}(B) \in \Sigma_2$.

We call this function $f$ measurable because we can define a measure $\mu: \Sigma \to \overline{\mathbb{R}}$ on $(\Omega_2, \Sigma_2)$ in terms of $(\Omega_1, \Sigma_1)$. Where $\mu$ has the following conditions:

1. $\mu(\varnothing) = 0$.
2. $\mu(B) \geq 0$ for all $B \in \Sigma$.
3. $\mu(\bigcup_{i \in I} B_i) = \sum_{i \in I} \mu(B_i)$ for all $\{B_i \in \Sigma\}_{i \in I}$ Countable Disjoint Family.

If $(\Omega, \Sigma)$ is a measurable space and $\mu$ is a measure, we call $(\Omega, \Sigma, \mu)$ a measure space, if $\mu$ is a probability measure i.e., $\mu(\Omega) = 1$, then $(\Omega, \Sigma, \mu)$ is a probability space.

Let $(\Omega_1, \Sigma_1, Pr)$ be a probability space and $(\Omega_2, \Sigma_2)$ be a measurable space. If $X: \Omega_1 \to \Omega_2$ is a measurable function, then $X$ is called a random variable with values in $(\Omega_2, \Sigma_2)$.

2.1.3. Kernel

Stochastic processes can model the process of generating one population from another population. A transition kernel is used to characterize each iteration in a stochastic process given by:

$$K(x, A) = P(x, A) = Pr[X_t \in A \mid X_{t-1} = x]. \tag{2}$$

where $x$ is the current population and $A$ a subset of possible populations.

**Definition 1.** *(**Markov kernel**) Let $(\Omega_1, \Sigma_1)$ and $(\Omega_2, \Sigma_2)$ be measurable spaces. A function $K: \Omega_1 \times \Sigma_2 \to [0, 1]$ is called a (Markov) kernel if the following two conditions hold:*

1. *Function $K_{x, \bullet}: A \mapsto K(x, A)$ is a probability measure for each fixed $x \in \Omega_1$*
2. *Function $K_{\bullet, A}: x \mapsto K(x, A)$ is a measurable function for each fixed $A \in \Sigma_2$.*

The transition kernel of an existing transition density $K: \Omega_1 \times \Omega_2 \rightarrow [0,1]$ is defined by:

$$K(x, A) = \int_A K(x, y) dy. \tag{3}$$

Kernels linked with deterministic methods used by SGOALs play a significant role in developing a systematic formal theory. Next, we review several characterizations of stochastic methods using transition kernels.

Deterministic Kernel

Let $(\Omega_1, \Sigma_1)$ and $(\Omega_2, \Sigma_2)$ be measurable spaces, and $f : \Omega_1 \rightarrow \Omega_2$ be $\Sigma_1 - \Sigma_2$ measurable. The function $1_f : \Omega_1 \times \Sigma_2 \rightarrow [0,1]$ is a kernel defined by:

$$1_f(x, A) = \begin{cases} 1 & \text{if } f(x) \in A \\ 0 & \text{otherwise.} \end{cases} \tag{4}$$

Kernel Indicator

Let $(\Omega, \Sigma)$ be a measurable space. The indicator function $1 : \Omega \times \Sigma \rightarrow [0,1]$ defined as $1(x, A) = 1_{id(x)}(A)$, with $id(x) = x$ is a kernel.

Random Scan (Mixing)

The mixing update mechanism of a set of $n$ Markov transition kernels $K_1, \ldots, K_n$, each of them with a probability of being picked $p_1, p_2, \ldots, p_n$ ($\sum p_i = 1$), is defined by:

$$\left( \sum_{i=1}^{n} p_i K_1 \right)(x, A) = \sum_{i=1}^{n} p_i \int_A K_i(x, y) dy. \tag{5}$$

Composition

The composition update mechanism is built using the kernel multiplication operator. It is built on the concept of applying one kernel after another. Kernel composition of $K_1, K_2$ is defined by:

$$(K_2 \circ K_1)(x, A) = \int K_2(y, A) K_1(x, dy). \tag{6}$$

The composition of update mechanisms that correspond to a set of $n$ transition kernels $K_1, \ldots, K_n$ is defined as the product kernel $K_n \circ K_{n-1} \circ \ldots \circ K_1$. Based on the fact that kernel multiplication is an associative operation.

Transition Kernel Iteration

The transition probability of iteration t (application) of a Markovian kernel $K$, describes the probability to transit to set $A \in \Sigma$ within $t$ steps when starting at state $x \in \Omega$ as defined by:

$$K^{(t)}(x, A) = \begin{cases} K(x, A) & , t = 1 \\ \int_\Omega K^{(t-1)}(y, A) K(x, dy) & , t > 1. \end{cases} \tag{7}$$

Let $p : \Sigma \rightarrow [0,1]$ be the initial distribution of subsets, in that case the probability that the Markov process is in set $A \in \Sigma$ at step $t \geq 0$ is given by:

$$Pr\{X_t \in A\} = \begin{cases} p(A) & , t = 0 \\ \int_\Omega K^{(t)}(x, A) p(dx) & , t > 0. \end{cases} \tag{8}$$

### 2.1.4. Kernels on Cartesian Products

SGOALswork with populations, so we need to work with probability theory concepts that can handle them; To do so, we use a probability space $\left(\Omega^n, \Sigma^{\otimes n}, \bigotimes_{i=1}^n \mu_i\right)$, where $\Omega^n$ is the set of elementary elements, i.e., the set of all possible populations of size n. $\Sigma^{\otimes n}$, is the product $\sigma$-algebra and $\bigotimes_{i=1}^n \mu_i$ a product probability measure [11].

We can use probability theory on cartesian products to use several new kernels by joining simple kernels. Hence, we can join stochastic methods used in SGOALsto generate new populations from other populations.

#### SWAP Kernel

We can define a kernel $1_{\leftarrow} : (\Omega_1 \times \Omega_2) \times (\Sigma_2 \otimes \Sigma_1) \to [0,1]$ that characterizes the Swap method, a deterministic method often utilized by SGOALs when we need to select individuals. Where $\leftarrow$ is defined as follows:

$$\leftarrow : \quad \begin{array}{ccc} \Omega_1 \times \Omega_2 & \to & \Omega_2 \times \Omega_1 \\ (x,y) & \mapsto & (y,x). \end{array} \tag{9}$$

#### PROJECTION Kernel

We can define a kernel $1_{\pi_I} : \prod_{i=1}^n \Omega_i \times \bigotimes_{i=1}^m \Sigma_i \to [0,1]$ that characterizes a method that selects m individuals from a population of size n. where $\pi_I$ is defined as follows:

$$\pi_I : \quad \begin{array}{ccc} \prod_{i=1}^n \Omega_i & \to & \prod_{i=1}^m \Omega_{k_i} \\ (x_1,\ldots,x_n) & \mapsto & (x_{k_1},\ldots,x_{k_n}). \end{array} \tag{10}$$

#### JOIN Kernel

The join of methods that generate a subpopulation of the next population can be characterized by a join kernel $\circledast K : \Omega^\eta \times \Sigma^{\otimes v} \to [0,1]$ with $v = \sum_{k=1}^m v_k$. Where the join stochastic method is defined as follows:

$$\mathrm{F} : \quad \begin{array}{ccc} \Omega^\eta & \to & \Omega^v \\ (x_1,\ldots,x_n) & \mapsto & \prod_{i=1}^m \mathrm{F}_i. \end{array} \tag{11}$$

#### PERMUTATION Kernel

Methods that generate permutations of populations can be characterized by the kernel $K_I : \Omega^n \times \Sigma^{\otimes n} \to [0,1]$ defined as $K_I = \circledast_{k=1}^n \pi_{i_k}$. Where $I = [i_1, i_2, \ldots, i_n]$ is a fixed permutation of the set $\{1, 2, \ldots, n\}$. If we let $\mathscr{P}$ as the set of permutations of the set $\{1, 2, \ldots, n\}$. then, the kernel $K_{\mathscr{P}} : \Omega^n \times \Sigma^{\otimes n} \to [0,1]$ defined as $K_{\mathscr{P}} = \frac{1}{|\mathscr{P}|} \sum_{I \in \mathscr{P}} \pi_I$ characterizes a stochastic method that generates a population from another one, looking for a fixed set of permutations.

#### SORTING Kernel

Methods that sorts populations according to closeness of objective function are commonly used in SGOALS, so we need to characterize them with a kernel $S_{n,n-1} : \Omega^n \times \Sigma^{\otimes n} \to [0,1]$. As defined in Proposition 63 in [11].

#### VR Kernel

A common pattern in most SGoals is that they can be described by two consecutive stochastic methods: A variation $\mathrm{V} : \Omega^\eta \to \Omega^\varpi$ and a replacement $\mathrm{R} : \Omega^{\eta+\varpi} \to \Omega^v$ method. Gomez in [4] defined these methods as a single variation-replacement $\mathrm{F} : \Omega^\eta \to \Omega^v$ method named (**VR**) method. These methods can be characterized by kernels $K_{\mathrm{R}} : \Omega^{\eta+\varpi} \times \Sigma^{\otimes v} \to [0,1]$ and $K_{\mathrm{V}} : \Omega^\eta \times \Sigma^{\otimes \varpi} \to [0,1]$, hence $K_{\mathrm{VR}} = K_{\mathrm{R}} \circ [1_{\Omega^\eta} \circledast K_{\mathrm{V}}]$ and $K_{\mathrm{VR}} = K_{\mathrm{R}} \circ [K_{\mathrm{V}} \circledast 1_{\Omega^\eta}]$.

### 2.2. Characterization of a SGOAL Using Probability Theory

**Definition 2.** *(optimality)* $d(x) = f(\text{BEST}(x)) - f^*$ *(Where $f^*$ is the optimal value of the objective function $f$ in $\Omega$).*

Gomez in [11], defines an optimization space using sets that include optimal elements. It can be seen that these sets can be related to the concept of level set. We establish optimal individuals as individuals that have an objective function less than $\epsilon \in \mathbb{R}^+$. The optimal elements are defined as follows.

1. ($\epsilon$-**state**) $x$ is $\epsilon$-optimum element if $d(x) < \epsilon$,
2. ($\bar{\epsilon}$-**state**) $x$ is an $\epsilon$-optimum element if $d(x) \leq \epsilon$, and
3. ($\hat{\epsilon}$-**state**) $x$ is an $\epsilon$-element if $d(x) = \epsilon$.

Gomez in [11] defines a **($f-$optimization $\sigma$-algebra)** as $\sigma$-algebra that contains strict $\epsilon$-optimum states $\{\Omega_\epsilon\}_{\epsilon>0} \subseteq \Sigma$. This in order to finally define an optimization space $(\Omega^n, \Sigma^{\otimes n}, f)$ to study the convergence of SGOALs. There $\Omega$ is the feasible region, $\Sigma$ is a $f$-optimization $\sigma$-algebra and $f$ is an objective function.

### 2.3. Kernels on Optimization Spaces
Elitist Stochastic Methods

Some SGOALS use elitist stochastic methods that guarantee that the best $v$ solutions in the next generation are equal to or better than the $v$ solutions of the current generation. It is to capture the notion of improving the solution.

**Definition 3.** *(elitist method)* A stochastic method $\text{F} : \Omega^\eta \to \Omega^v$ is called elitist if $f(\text{BEST}(\text{F}(P))) \leq f(\text{BEST}(P))$.

**Definition 4.** *(elitist kernel)* A kernel $K : \Omega^\eta \times \Sigma^{\otimes v} \to [0, 1]$ is called elitist if $K(x, A) = 0$ for each $A \in \Sigma^{\otimes v}$ such that $d(x) < d(y)$ for all $y \in A$.

**Lemma 1.** *If $K : \Omega^\eta \times \Sigma^{\otimes v} \to [0, 1]$ is elitist then*

1. $K\left(x, \left(\Omega^v_{d(x)}\right)^c\right) = 0$ *and* $K\left(x, \Omega^v_{d(x)}\right) = 1.$
2. *Let $x \in \Omega^\eta$, if $d(x) < \alpha \in \mathbb{R}$ then $K\left(x, \left(\Omega^v_\alpha\right)^c\right) = 0$ and $K\left(x, \Omega^v_\alpha\right) = 1.$*

**Definition 5.** *(optimal strictly bounded from zero)* A kernel $K : \Omega^\eta \times \Sigma^{\otimes v} \to [0, 1]$ is called optimal strictly bounded from zero iff $K(x, \Omega_\epsilon) \geq \delta(\epsilon) > 0$ for all $\epsilon > 0$.

### 2.4. Convergence of a SGoal
2.4.1. Convergence

Let $(D_t)$ be a random sequence, i.e., a sequence of random variables defined on a probability space $(\Omega, \Sigma, P)$. Then $(D_t)$ is said to

1. converge completely to zero , denoted as $D_t \xrightarrow{c} 0$, if for every $\epsilon > 0$

$$\lim_{t \to \infty} \sum_{i=1}^{t} Pr\{|D_t| > \epsilon\} < \infty. \tag{12}$$

2. converge in probability to zero, denoted as $D_t \xrightarrow{p} 0$, if for every $\epsilon > 0$

$$\lim_{t \to \infty} Pr\{|D_t| > \epsilon\} = 0. \tag{13}$$

Gomez in [11] follows the approach proposed by Günter Rudolph in [24], to study the convergence properties of a SGOAL. This concept is also studied by Zhigljavsky and Zilinskas in [6] , where there is an extensive study of SGOALs. We clarify that in the rest of

this paper, $\Sigma$ is an optimization $\sigma$-algebra. First, Rudolph defines a convergence property for a SGOAL in terms of the objective function.

**Definition 6.** (**SGOAL** *convergence*). *Let $P_t \in \Omega^n$ be the population maintained by a SGOAL $\mathcal{A}$ at iteration t. Then $\mathcal{A}$ converges to the global optimum if the random sequence $(D_t = d(P_t) : t \geq 0)$ converges completely to zero.*

**Lemma 2.** *(Lemma 1 in [24]) If $\exists \epsilon > 0$ such that $K(x, \Omega_\epsilon) \geq \delta > 0$ for all $x \in \Omega_\epsilon^c$ and $K(x, \Omega_\epsilon) = 1$ for all $x \in \Omega_\epsilon$ then, holds for $t \geq 1$*

$$K^{(t)}(x, \Omega_\epsilon) \geq 1 - (1 - \delta)^t. \tag{14}$$

Using Lemma 2, Rudolph establishes a theorem for convergence of evolutionary algorithms. Which is specified towards SGOALsin Theorem 1.

**Theorem 1.** *(Theorem 1 in Rudolph [24])*

*Let a SGOAL fulfill the condition everywhere dense sampling condition of Lemma 2. Then will converge to the global optimum ($f^*$) of a real valued function $f : \Phi \to \mathbb{R}$ with $f > -\infty$, defined in an arbitrary space $\Omega \subseteq \Phi$, regardless of the initial distribution $p(\cdot)$.*

### 2.4.2. Convergence of a VR-SGOAL

Gomez [11] follows the approach used by Günter Rudolph in [24], to study the convergence properties of a VR-SGOALs but Gomez rewrites it in terms of kernels VR variaton-replacement.

**Theorem 2.** *A VR-SGOAL with $K_V$ an optimal strictly bounded from zero variation kernel and $K_R$ an elitist replacement kernel, will converge to the global optimum of the objective function.*

## 3. Materials and Methods

In this section, we begin by generalizing the theory developed by Gomez in [11] to generalize the concept of Stationary Markov process to **non-stationary** ones. Next, we study the necessary theory to characterize arithmetic methods that are useful in some recombination and mutation schemes described in [25].

### 3.1. Generalization to Non-Stationary Algorithms

For a non-stationary (or non-homogeneous) Markov process, the transition probabilities (kernel) may change over time ([26]). Suppose that $K_t$ is the transition kernel applied at time $t > 0$ of a non-stationary Markov process. Then, the transition kernel of such non-stationary Markov process at time $t$ is defined as $K^{(t)} = K_t \circ K_{t-1} \circ \ldots \circ K_1$. Clearly, we can rewrite Equation (7) . The transition kernel of a non-stationary Markov process is given by:

$$K^{(t)}(x, A) = \begin{cases} K^1(x, A) & \text{if } t = 1 \\ \int_\Omega K^{(t-1)}(y, A) K^t(x, dy) & \text{if } t > 1. \end{cases} \tag{15}$$

Now we are in the position of generalizing Lemma 71 in [11] to non-stationary Markov processes.

**Lemma 3.** *If $\exists \delta > 0$, such that for all $x \in \Omega_\epsilon^c$, $K_t(x, \Omega_\epsilon) \geq \delta > 0$ and, for all $x \in \Omega_\epsilon$, $K_t(x, \Omega_\epsilon) = 1$, then $K^{(t)}(x, \Omega_\epsilon) \geq 1 - (1 - \delta)^t$ holds for $t \geq 1$.*

**Proof.** We just rewrite the proof of Lemma 71 in [11] (Gomez uses induction on $t$) but taking care of the non-stationary property of the Markov process. For $t = 1$ we have that

$K^{(t)}(x, \Omega_\epsilon) = K_t(x, \Omega_\epsilon)$ (Equation (7)), so $K^{(t)}(x, \Omega_\epsilon) \geq \delta$ (condition lemma), therefore $K^{(t)}(x, \Omega_\epsilon) \geq 1 - (1 - \delta)^t$ ($t = 1$ and numeric operations). Here, we will use the notation (as Gomez did) $K^{(t)}(y, \Omega_\epsilon) = K_y^{(t)}(\Omega_\epsilon)$ to reduce the visual length of the equations.

$$
\begin{aligned}
&K_x^{(t+1)}(\Omega_\epsilon) \\
&= \int_\Omega K_y^{(t)}(\Omega_\epsilon) K_t(x, dy) && \text{(Equation (7))} \\
&= \int_{\Omega_\epsilon} K_y^{(t)}(\Omega_\epsilon) K_t(x, dy) + \int_{\Omega_\epsilon^c} K_y^{(t)}(\Omega_\epsilon) K_t(x, dy) && (\Omega = \Omega_\epsilon \cup \Omega_\epsilon^c) \\
&= \int_{\Omega_\epsilon} K_t(x, dy) + \int_{\Omega_\epsilon^c} K_y^{(t)}(\Omega_\epsilon) K_t(x, dy) && (\text{If } y \in \Omega_\epsilon, K_y^{(t)}(\Omega_\epsilon) = 1) \\
&= K_t(x, \Omega_\epsilon) + \int_{\Omega_\epsilon^c} K_y^{(t)}(\Omega_\epsilon) K_t(x, dy) && \text{(def kernel)} \\
&\geq K_t(x, \Omega_\epsilon) + \left[1 - (1 - \delta)^t\right] \int_{A_\epsilon^c} K_t(x, dy) && \text{(Induction hypothesis)} \\
&\geq K_t(x, \Omega_\epsilon) + \left[1 - (1 - \delta)^t\right] K_t(x, \Omega_\epsilon^c) && \text{(def kernel)} \\
&\geq K_t(x, \Omega_\epsilon) + K_t(x, \Omega_\epsilon^c) - (1 - \delta)^t K_t(x, \Omega_\epsilon^c) \\
&\geq 1 - (1 - \delta)^t (1 - K_t(x, \Omega_\epsilon)) && \text{(Probability)} \\
&\geq 1 - (1 - \delta)^t (1 - \delta) && \text{(condition lemma)} \\
&\geq 1 - (1 - \delta)^{t+1}.
\end{aligned}
$$

$\square$

Finally, Theorem 72 in [11] also holds for non-stationary Markov processes. So, in order to show convergence of a non-stationary SGOAL it is sufficient to prove that the SGOAL satisfies the condition of Lemma 3.

**Theorem 3.** *(Theorem 72 in [11]—a corrected version of Theorem 1 in [24]) A* SGOAL *whose stochastic kernel satisfies $K^{(t)}(x, \Omega_\epsilon) \geq 1 - (1 - \delta)^t$ for all $t \geq 1$ will converge to the global optimum ($f^*$) of a well-defined real-valued function $f : \Phi \to \mathbb{R}$, defined in an arbitrary space $\Omega \subseteq \Phi$, regardless of the initial distribution $p(\cdot)$.*

**Proof.** See proof of Theorem 72 in [11]. $\square$

*3.2. Arithmetic between Measurable Functions*

Arithmetic operations can be found in schemes of mutation and recombination [25]. So to characterize an algorithm with kernels in its entirety we must characterize all methods that can alter the generation of new populations.

According to Theorem 22 in [11], to characterize arithmetic methods as deterministic kernels it is enough to prove that these methods are measurable. Proposition 1 provides the sufficient conditions for a function $f : \Omega \to \mathbb{R}$ to be measurable.

**Proposition 1.** *Let $(\Omega, \Sigma)$ be a measurable space, then $f : \Omega \to \mathbb{R}$ is $\Sigma - \mathcal{B}(\mathbb{R})$ measurable if and only if one of the following conditions holds:*

1. $\{x \in \Omega : f(x) < b\} \in \Sigma$ for every $b \in \mathbb{R}$
2. $\{x \in \Omega : f(x) \leq b\} \in \Sigma$ for every $b \in \mathbb{R}$
3. $\{x \in \Omega : f(x) \geq b\} \in \Sigma$ for every $b \in \mathbb{R}$
4. $\{x \in \Omega : f(x) > b\} \in \Sigma$ for every $b \in \mathbb{R}$.

**Proof.** Note that $\{x \in \Omega : f(x) < b\} = f^{-1}((-\infty, b))$, and use $\{(-\infty, b) : b \in \mathbb{R}\}$ family to generate $\mathcal{B}(\mathbb{R})$. Details see [27] proposition 1 chapter 3. $\square$

Lemma 4 gives a useful equality between sets to characterize arithmetic methods.

**Lemma 4.** *Let $(\Omega, \Sigma)$, be a measurable space and $f\colon \Omega \to \mathbb{R}$ and $g\colon \Omega \to \mathbb{R}$ be $\Sigma - \mathscr{B}(\mathbb{R})$ measurable, then*

$$\{(x,y) \in \Omega \times \Omega \colon f(x) + g(y) < c\} = \bigcup_{q \in \mathbb{Q}} L_{q,c} \times R_{q,c}, \tag{16}$$

*where*

$$L_{q,c} = \{x \in \Omega \colon g(x) < c - q\}$$

*and*

$$R_{q,c} = \{y \in \Omega \colon f(y) < q\}.$$

**Proof.** $[\supseteq]$ Consider a tuple $(x,y)$ such that:

$$(x,y) \in \bigcup_{q \in \mathbb{Q}} L_{q,c} \times R_{q,c}$$

then, for some $q \in \mathbb{Q}$ we have that $g(x) < c - q$ and $f(y) < q$. Hence, applying an arithmetic operation we have that $f(y) + g(x) < c$. So $(x,y) \in \{(x,y) \in \Omega \colon f(y) + g(x) < c\}$.

$[\subseteq]$

Let $(x,y) \in \{(x,y) \in \Omega \colon f(y) + g(x) < c\}$ so there exists some $q \in \mathbb{Q}$ that by the density of rational numbers holds: $f(y) < q < c - g(x)$ applying some arithmetic $f(y) < q$ and $g(x) < c - q$. From,

$$(x,y) \in \bigcup_{q \in \mathbb{Q}} [\{x \in \Omega \colon g(x) < c - q\} \times \{y \in \Omega \colon f(y) < q\}$$

follows that

$$\{(x,y) \in \Omega \times \Omega \colon f(x) + g(y) < c\} \subseteq \bigcup_{q \in \mathbb{Q}} L_{q,c} \times R_{q,c}.$$

$\square$

3.2.1. Method Product by a Scalar

**Proposition 2.** *Let $(\Omega, \Sigma)$, be a measurable space and $f\colon \Omega \to \mathbb{R}$ be $\Sigma - \mathscr{B}(\mathbb{R})$ measurable, then $h\colon \Omega \to \mathbb{R}$ defined as $h(x) = \alpha * f(x)$ where $\alpha \in \mathbb{R}$, is $\Sigma - \mathscr{B}(\mathbb{R})$ measurable.*

**Proof.** For every $\alpha \in \mathbb{R}$ and $c \in \mathbb{R}$ we want to show that $h^{-1} \in \Sigma$, we prove by cases.

$[\alpha = 0]$      Note that $h^{-1}((-\infty, c)) = \{x \in \Omega \colon 0 * f(x) = 0\}$
     $\{x \in \Omega \colon 0 * f(x) = 0\} = \Omega$      Because $f(x) \in \mathbb{R}$
     $\Omega \in \Sigma$      Definition of $\sigma$-algebra
     $h^{-1}(\{0\}) \in \Sigma$      $h^{-1}(\{0\}) = \{x \in \Omega \colon 0 * f(x) = 0\}$

$[\alpha > 0]$      Note that $h^{-1}((c, \infty)) = \{x \in \Omega \colon \alpha * f(x) > c\}$
     $h^{-1}((c, \infty)) = \{x \in \Omega \colon f(x) > c/\alpha\}$      Arithmetic operations
     $\{x \in \Omega \colon f(x) > c/\alpha\} \in \Sigma$      Measurable by proposition 2.
     $h^{-1}(c, \infty) \in \Sigma$      $h^{-1}(c, \infty) = \{x \in \Omega \colon \alpha * f(x) > c\}$

$[\alpha < 0]$      Note that $h^{-1}((-\infty, c)) = \{x \in \Omega \colon \alpha * f(x) < c\}$
     $h^{-1}((-\infty, c)) = \{x \in \Omega \colon f(x) < c/\alpha\}$      Arithmetic operations
     $\{x \in \Omega \colon f(x) < c/\alpha\} \in \Sigma$      Measurable by proposition 2.
     $h^{-1}((-\infty, c)) \in \Sigma$      $h^{-1}((-\infty, c)) = \{x \in \Omega \colon \alpha * f(x) < c\}$.

$\square$

### 3.2.2. Method Addition

**Proposition 3.** *Let $(\Omega, \Sigma)$ be a measurable space and $f: \Omega \to \mathbb{R}$ and $g: \Omega \to \mathbb{R}$ be $\Sigma - \mathscr{B}(\mathbb{R})$ measurable functions, then $h: \Omega \times \Omega \to \mathbb{R}$ defined as $h(x,y): f(y) + g(x)$ is $\Sigma \otimes \Sigma - \mathscr{B}(\mathbb{R})$ measurable.*

**Proof.** We want to show that $h^{-1}((-\infty, c)) \in \Sigma \otimes \Sigma$ for all $c \in \mathbb{R}$ according to proposition 2. Now, note that

$$\{(x,y): f(y) + g(x) < c\} = h^{-1}((-\infty, c))$$

And using Lemma 4 we establish that:

$$\{(x,y) \in \Omega \times \Omega: g(x) + f(y) < c\} = \bigcup_{q \in \mathbb{Q}} L_{q,c} \times R_{q,c}$$

where

$$L_{q,c} = \{x \in \Omega: g(x) < c - q\}$$

and

$$R_{q,c} = \{y \in \Omega: f(y) < q\}$$

So, if we show that $\bigcup_{q \in \mathbb{Q}} L_{q,c} \times R_{q,c} \in \Sigma \otimes \Sigma$, then $h$ is measurable.

| | |
|---|---|
| $\{x \in \Omega: g(x) < c - q\} \in \Sigma$ | measurable by proposition 2 |
| $\{y \in \Omega: f(y) < q\} \in \Sigma$ | measurable by proposition 2 |
| $L_{q,c} \times R_{q,c} \in \Sigma \times \Sigma$ | family product definition in [11] |
| $L_{q,c} \times R_{q,c} \in \Sigma \otimes \Sigma$ | $\Sigma \times \Sigma \subseteq \Sigma \otimes \Sigma$. |
| $\bigcup_{q \in \mathbb{Q}} L_{q,c} \times R_{q,c} \in \Sigma \otimes \Sigma$ | $\Sigma \otimes \Sigma$ is $\overline{CU}$ |
| So $h^{-1}((-\infty, c)) \in \Sigma \otimes \Sigma$ | $h^{-1}((-\infty, c)) = \bigcup_{q \in \mathbb{Q}} L_{q,c} \times R_{q,c}$. |

$\square$

### 3.2.3. Method Product

**Lemma 5.** *Let $(\Omega, \Sigma)$ be a measurable space and $f: \Omega \to \mathbb{R}$ be $\Sigma - \mathscr{B}(\mathbb{R})$ measurable function, then $h: \Omega \to \mathbb{R}$ defined as $h(x): f^2(x)$ is $\Sigma - \mathscr{B}(\mathbb{R})$ measurable.*

**Proof.** We need to show that $h(x)$ is measurable. i.e., we need to show that $h^{-1}((c, \infty)) \in \Sigma$, for all $c \in \mathbb{R}$, note that $h^{-1}((c, \infty)) = \{x \in \Omega: f^2(x) > c\}$, We prove by cases.

| | |
|---|---|
| $[c \geq 0]$ | |
| $\{x \in \Omega: f^2(x) > c\} =$ | . |
| $\{x \in \Omega: f(x) > \sqrt{c}\} \cup \{x \in \Omega: f(x) < -\sqrt{c}\}$ | Inequality |
| $\{x \in \Omega: f(x) > \sqrt{c}\} \in \Sigma$ | Measurable by proposition 2 |
| $\{x \in \Omega: f(x) < -\sqrt{c}\} \in \Sigma$ | Measurable by proposition 2 |
| $\{x \in \Omega: f(x) > \sqrt{c}\} \cup \{x \in \Omega: f(x) < -\sqrt{c}\} \in \Sigma$ | $\Sigma$ is $\overline{CU}$ |
| $[c < 0]$ | |
| $\{x \in \Omega: f^2(x) > c\} = \Omega$ | All values of $f^2(x)$ are positive. |
| $\Omega \in \Sigma$ | Definition of $\sigma-$algebra |
| $h^{-1}(c, \infty) \in \Sigma$ | $\{x \in \Omega: f^2(x) > c\} = h^{-1}(c, \infty)$. |

$\square$

**Proposition 4.** *Let $(\Omega, \Sigma)$ be a measurable space and $f: \Omega \to \mathbb{R}$ and $g: \Omega \to \mathbb{R}$ be $\Sigma - \mathscr{B}(\mathbb{R})$ measurable functions, then $h: \Omega \times \Omega \to \mathbb{R}$ defined as $h(x,y): f(y) * g(x)$ is $\Sigma \otimes \Sigma - \mathscr{B}(\mathbb{R})$ measurable.*

**Proof.** We can observe that the product of two functions can be expressed as follows:

$$f(y) * g(x) = \tfrac{1}{2}[(f(y) + g(x))^2 - f(y)^2 - g(x)^2]$$

This is measurable because it is in terms of addition, square of measurable functions, and multiplication with a scalar. Follows from Propositions 2 and 3, and Lemma 5. □

### 3.2.4. Arithmetic Kernels

Now we proceed to characterize the arithmetic methods using deterministic kernels as defined in Section 2.1.3 that establishes as condition to work on a measurable function.

**Theorem 4.** *(**Addition kernel**) Let $h(x, y) = f(x) + g(y)$ be a measurable function as defined in Proposition 3. The addition function $\mathbf{1}_+\colon (\Omega \times \Omega) \times \mathscr{B}(\mathbb{R}) \to [1, 0]$ defined as $\mathbf{1}_h((x, y), A) = 1_{h(x,y)}(A)$ is a kernel.*

**Proof.** This is a deterministic kernel as defined in in Section 2.1.3, it is sufficient to prove that $h(x, y)$ is measurable, which is done in Proposition 3. □

**Theorem 5.** *(**Product-Scalar kernel**) Let $h(x) = \alpha * f(x)$ be a measurable function as defined in Proposition 2. The product-scalar function $\mathbf{1}_\alpha\colon \Omega \times \mathscr{B}(\mathbb{R}) \to [1, 0]$ defined as $\mathbf{1}_h(x, A) = 1_h(A)$ is a kernel.*

**Proof.** This is a deterministic kernel as defined in in Section 2.1.3, it is sufficient to prove that $h(x)$ is measurable, which is done in Proposition 2. □

### 3.2.5. Product Kernel

**Theorem 6.** *(**Product kernel**) Let $h(x, y) = f(x) * g(y)$ be a measurable function as defined in Proposition 4. The product function $\mathbf{1}_*\colon (\Omega \times \Omega) \times \mathscr{B}(\mathbb{R}) \to [1, 0]$ defined as $\mathbf{1}_h((x, y), A) = 1_{h(x,y)}(A)$ is a kernel.*

**Proof.** This is a deterministic kernel as defined in Section 2.1.3, it is sufficient to prove that $h(x, y)$ is measurable, which is done in Proposition 4. □

**Remark 1.** *For the sake of writing simplicity, in the rest of this paper, whenever we refer to an arithmetic kernel or a combination of these, we will use the symbol $\mathbf{1}_+$.*

## 4. Results

### 4.1. Selection Scheme Formalization

A Selection Scheme , is a method of selecting a group of individuals from a population [28]. Some studies of these schemes can be found in [29–31]. Many schemes define an individual selection mechanism $\text{S}1\colon \Omega^\lambda \to \Omega$, and selects a group of individuals by repeatedly applying $\text{S}1$. In this paper, we study the uniform, fitness proportional, tournament ([32]), roulette, and ranking selection schemes:

1. A uniform scheme ($\text{UNIFORM}1\colon \Omega^\lambda \to \Omega$) gives to each candidate solution $i = 1, 2, \ldots, \lambda$, the same selection probability $p(x_i) = \frac{1}{\lambda}$.
2. A fitness proportional scheme ($\text{PROPORTIONAL}1\colon \Omega^\lambda \to \Omega$) gives to each candidate solution $i = 1, 2, \ldots, \lambda$, a selection's probability $p(x_i)$ such that $p(x_i) < p(x_j)$ if $f(x_j) \lhd f(x_i)$ and $p(x_i) = p(x_j)$ if $f(x_i) = f(x_j)$.
3. A tournament scheme ($\text{TOURNAMENT}1^m\colon \Omega^\lambda \to \Omega$) of size $m$ chooses $m$ individuals using a $\text{UNIFORM}$ scheme and selects an individual from these using a $\text{PROPORTIONAL}1$ scheme, $\text{TOURNAMENT}1^m = \text{PROPORTIONAL}1 \circ \text{UNIFORM}^m$.
4. A roulette scheme ($\text{ROULETTE}1\colon \Omega^\lambda \to \Omega$) is a fitness proportional one where $p(x_i) = \frac{rate(x_i)}{\sum_{i=1}^\lambda rate(x_i)}$ with $rate(x_i) < rate(x_j)$ if $f(x_j) \lhd f(x_i)$ and $rate(x_i) = rate(x_j)$ if $f(x_i) = f(x_j)$. If $f(x_i) \geq 0$ for all $i = 1, 2, \ldots, \lambda$ and maximizing then $rate(x_i)$ can be set to $f(x_i)$.
5. A ranking scheme ($\text{RANKING}1\colon \Omega^\lambda \to \Omega$) is a roulette one with

$$rate(x_i) = 1 + |\{x_k \colon f(x_i) \lhd f(x_k)\}|.$$

6. A stud scheme (STUD1: $\Omega^\lambda \to \Omega$) chooses the best candidate and can be characterized for the next kernel $K_{R_{\mu,\mu+\lambda}} = \pi_{\{1\}} \circ s_{\lambda,\lambda-1}$

7. An Over Selection scheme as defined in [33] (OSELECTION1: $\Omega^\lambda \to \Omega$) is a roulette one with $p(x_i) = 0.8/(\lambda * 0.68)$ if $ranking(x_i) \lhd \lambda * 0.68$ and $p(x_i) = 0.2/(\lambda * 0.32)$ if $ranking(x_i) \rhd \lambda * 0.32$ where $ranking(x_i)$ is defined as
$ranking(x_i) = 1 + |\{x_k \colon f(x_i) \lhd f(x_k)\}|.$

**Proposition 5.** *If* S1: $\Omega^\lambda \to \Omega$ *is a selection scheme with kernel $K_{S1}$ then* S: $\Omega^\lambda \to \Omega^\mu$ *has kernel $K_S = \circledast_{i=1}^\mu K_{S1}$.*

**Corollary 1.** *If* S1 *is based on a probability function then $K_S$ is a kernel.*

**Corollary 2.** *The* UNIFORM, PROPORTIONAL, TOURNAMENT, ROULETTE, RANKING, STUD *and* OSELECTION *selection schemes have Markov kernels.*

*4.2. Recombination Scheme Formalization*

Recombination schemes use information from one or more parents and generate offspring that share information with their parents. Details for each scheme can be found in [25,34–36].

In the following characterizations each individual from a population belongs to $\Omega^n$ (set of elementary events), where $n$ is the dimension. Keep in mind that all theory developed by [11] is applicable, hence it is generalized from tuples of tuples to a single tuple. Review Proposition 32 and Corollary 33 in [11].

1. A Single-Point Crossover method (SPC1$_d$: $\Omega^n \times \Omega^n \to \Omega^n \times \Omega^n$) is described in Algorithm 2 and can be characterized by the next kernel:

$$K_{SPC_{Q_l}} = \pi_{\{1...d\}}(A) \circledast \pi_{\{d+1...n\}}(B)$$

$$K_{SPC_{Q_r}} = \pi_{\{1...d\}}(B) \circledast \pi_{\{d+1...n\}}(A)$$

$$K_{SPC} = K_{SPC_{Q_l}} \circledast K_{SPC_{Q_r}}. \tag{17}$$

**Proof.** $K_{SPC}$ is defined in terms of projection kernel and join-kernels. $\square$

---

**Algorithm 2** Single Point Crossover-SPC1

---

SPC1$_d(A, B)$

1: $A_l = \pi_{\{1,...,d\}}(A)$
2: $A_r = \pi_{\{d+1,...,n\}}(A)$
3: $B_l = \pi_{\{1,...,d\}}(B)$
4: $B_r = \pi_{\{d+1,...,n\}}(B)$
5: $Q_l = \text{JOIN}(A_l, B_r)$
6: $Q_r = \text{JOIN}(B_l, A_r)$
7: **return** $(Q_l, Q_r)$

---

2. A Multiple-Point Crossover scheme (MPC1$_D$: $\Omega^n \times \Omega^n \to \Omega^n \times \Omega^n$). Let $D = \{1, d_1, d_2, \ldots d_m, n\}$ be an ordered list of $\{m + 2 \in \mathbb{N}^+\}$ integers that indicate the m positions of crossover plus the first and last position. This formalization just considers when m is an odd number. We can see in Algorithm 3 the description of the algorithm and can be characterized by the next kernel. where $l$ is the length of $D$.

$$K_{MPC_{Q_l}} = \circledast_{i=1}^{l/2}(\pi_{\{D_{i*2-1}...D_{i*2}\}}(A) \circledast \pi_{\{D_{i*2+1}...D_{i*2+1}\}}(B))$$

$$K_{MPC_{Q_r}} = \circledast_{i=1}^{l/2}(\pi_{\{D_{i*2-1}...D_{i*2}\}}(B) \circledast \pi_{\{D_{i*2+1}...D_{i*2+1}\}}(A))$$

$$K_{MPC} = K_{MPC_{Q_l}} \circledast K_{MPC_{Q_r}}. \tag{18}$$

**Proof.**     $K_{MPC}$ is defined in terms of projection kernel and join-kernels.     □

---

**Algorithm 3** Multiple Point Crossover-MULTIPLEPOINT1

---

1: $D = [1, d_1, d_2, \ldots d_m, n]$

MULTIPLEPOINT1$_D$ $(A, B)$

1: $Q_l = \{\}$
2: $Q_r = \{\}$
3: **for** $i = 1$ to $length(D)/2$ **do**
4:     $Q_{l_i}$ = JOIN$(\pi_{\{D_{i*2-1},...,D_{i*2}\}}(A), \pi_{\{D_{i*2+1},...,D_{i*2+1}\}}(B))$
5: **for** $i = 2$ to $length(D)/2$ **do**
6:     $Q_{r_i}$ = JOIN$(\pi_{\{D_{i*2-1},...,D_{i*2}\}}(B), \pi_{\{D_{i*2+1},...,D_{i*2+1}\}}(A))$
7: **return** $(Q_l, Q_r)$

---

3.     A **Multi-Parent Crossover** scheme (MULTIPARENTC1: $\Omega^{n*b} \to \Omega^n$) can be considered as a generalization of **Uniform Crossover**, Where the definition is given in Algorithm 4. There, the method (SIZE: $\Omega^{n*b} \to \mathbb{N} \times \mathbb{N}$) in line 1 calculates the number of features of each individual and the amount of parents $(n, b)$ respectively. The method GENERATELISTINDEX1: $\mathbb{N} \times \mathbb{N} \to \mathbb{N}^n$ creates a list of length $n$ where each position has an integer that indicates some parent. This assignation is done following some rule defined in the design of the algorithm. Finally, the method CROSSOVER1$_D$: $\Omega^{n*b} \to \Omega^n$ assigns each element from the parents to a new individual according to values of $D$. In this characterization we can see that $P \in \Omega^{n*b}$, we are using this representation in order to use all theory created in [11] Section 3 that allow us to move from tuples of tuples to a single tuple.

The method CROSSOVER1$(D, n)$ as defined in Algorithm 4 can be characterized by a kernel $K_{Crossover}: \Omega^{n*b} \times \Sigma^{\otimes n} \to [0, 1]$ defined as:

$$K_{Crossover1} = \circledast_{i=1}^{n} \pi_{\{n*(D_i-1)+i\}}. \tag{19}$$

**Proof.**     $K_{Crossover1}$ is defined in terms of projection kernel and join kernels.     □

4.     A **Shuffle Crossover** scheme (SHUFFLEC1: $\Omega^n \times \Omega^n \to \Omega^n \times \Omega^n$). We start by permuting each parent. Next, we use some scheme that we have studied above to obtain children. Finally, we undo the permutation that we did to the beginning of the method. The definition can be seen in Algorithm 5. Where method RANDONPERMUTATION: $\mathbb{N} \to \mathbb{N}^n$ generates a permutation of a set of indexes corresponding to the length of features of each parent; CONVPERMUTATION$_{L_{per}}$: $\Omega^n \to \Omega^n$ sorts the features of the parents according to the set of indexes obtained in RANDONPERMUTATION; SEGMENTED1$_p$: $\Omega^n \times \Omega^n \to \Omega^n \times \Omega^n$ is the same as definition above; and CONVPERMUTATIONINV$_{L_{per}}$: $\Omega^n \to \Omega^n$ undo the permutation obtained after obtain the child. This method can be characterized by the next kernels:

$$K_{RandomPermutation} = K_{\mathscr{P}}$$

$$K_{ConvPermutation} = \circledast_{i=1}^{n} \pi_{L_{per_i}}$$

---

**Algorithm 4** MultiParent Crossover-MULTIPARENTC1

---

CROSSOVER1$_{D,n}$ $(P)$

1: $Q = \{\}$
2: **for** $i = 1$ to $n$ **do**
3:     $Q_i = \pi_{\{(n*(D_i-1))+i\}}(P)$
4: **return** $(Q)$

MULTIPARENTC1 $(P)$

1: n, b= SIZE(P)
2: D = GENERATELISTINDEX(n,b)
3: $Q =$ CROSSOVER1$_{D,n}$ $(P)$
4: **return** $(Q)$

---

$$K_{Segmented1_p} = K_{SC}$$

$$K_{ConvPermutationInv} = \circledast_{i=1}^{n} \pi_i$$

$$K_{\textbf{ShuffleC}} = [\circledast_{i=1}^{n} \pi_i \circledast \circledast_{i=1}^{n} \pi_i] \circ K_{SC} \circ [\circledast_{i=1}^{n} \pi_{L_{per_i}} \circledast \circledast_{i=1}^{n} \pi_{L_{per_i}}] \circ K_{\mathscr{P}}. \quad (20)$$

**Proof.**   $K_{\textbf{ShuffleC}}$ is defined in terms of projection kernel, join-kernels and kernel composition.   □

---

**Algorithm 5** Shuffle Crossover-SHUFFLEC1

---

CONVPERMUTATION1$_{L_{per}}$ $(P)$

1: $Q = \{\}$
2: **for** $i = 1$ to $n$ **do**
3:     $Q_i = \pi_{\{L_{per_i}\}}(P)$
4: **return** $(Q)$

CONVPERMUTATIONINV1$_{L_{per}}$ $(P)$

1: $Q = \{\}$
2: **for** $i = 1$ to $n$ **do**
3:     $Q_{\{L_{per_i}\}} = \pi_i(P)$
4: **return** $(Q)$

SHUFFLEC1$(A, B)$

1: $L_{per} =$ RANDOMPERMUTATION$(n)$
2: $A_{per} =$ CONVPERMUTATION$_{L_{per}}(A)$
3: $B_{per} =$ CONVPERMUTATION$_{L_{per}}(B)$
4: $Q_{1_{per}}, Q_{2_{per}} =$ SEGMENTED1$_p$ $(A_{per}, B_{per})$
5: $Q_1 =$ CONVPERMUTATIONINV$_{L_{per}}(Q_{1_{per}})$
6: $Q_2 =$ CONVPERMUTATIONINV$_{L_{per}}(Q_{2_{per}})$
7: **return** $(Q_1, Q_2)$

---

5.   **Flat Crossover or Arithmetic Crossover** schemes, we can use them when the features are defined in the real numbers. (FLATC1: $\Omega^n \times \Omega^n \to \Omega^n$), (ARITHMETICC1: $\Omega^n \times$

$\Omega^n \to \Omega^n \times \Omega^n$).The definitions can be seen in Algorithm 6. These methods can be characterized by the kernels $K_{FlatC1} \colon (\Omega^n \times \Omega^n) \times \Sigma^{\otimes n} \to [1,0]$, $K_{ArithmeticC1} \colon (\Omega^n \times \Omega^n) \times (\Sigma^{\otimes n} \times \Sigma^{\otimes n}) \to [1,0]$ where the definitions are:

$$K_{FlatC1} = \circledast_{i=1}^n [\mathbf{1}_{+_1} \circ [\pi_i(A) \circledast \pi_i(B)]]$$

$$K_{ArithmeticC1} = [\circledast_{i=1}^n [\mathbf{1}_{+_1} \circ [\pi_i(A) \circledast \pi_i(B)]] \circledast \circledast_{i=1}^n [\mathbf{1}_{+_2} \circ [\pi_i(A) \circledast \pi_i(B)]]]. \quad (21)$$

**Proof.** $K_{ArithmeticC1}$ is defined in terms of projection kernel, join-kernels and kernel composition. $\square$

---

**Algorithm 6** Flat and Arithmetic Crossover-SHUFFLEC1, ARITHMETICC1

---

FLATC1$(A, B)$

1: $Q = \{\}$
2: **for** $i = 1$ to $n$ **do**
3: $\quad \alpha \sim U[0,1]$
4: $\quad Q_i = \alpha * \pi_i(A) + (1 - \alpha) * \pi_i(B)$
5: **return** $(Q)$

ARITHMETICC $(A, B)$

1: $Q_l = \{\}$
2: $Q_r = \{\}$
3: **for** $i = 1$ to $n$ **do**
4: $\quad \alpha \sim U[0,1]$
5: $\quad Q_{l_i} = \alpha * \pi_i(A) + (1 - \alpha) * \pi_i(B)$
6: $\quad Q_{r_i} = (\alpha - 1) * \pi_i(A) + (\alpha) * \pi_i(B)$
7: **return** $(Q_l, Q_r)$

---

6. A **Blended Crossover** scheme, can be seen as a generalization of FLATC1. The scheme is represented by the function (BLENDEDC1$_\alpha \colon \Omega^n \times \Omega^n \to \Omega^n$).The definitions can be seen in Algorithm 7. These methods can be characterized by the kernels $K_{BlendedC1} \colon (\Omega^n \times \Omega^n) \times \Sigma^{\otimes n} \to [1,0]$, defined by:

$$K_{min_i} = \pi_1 \circ s_{2_\triangleleft} \circ [\pi_i(A) \circledast \pi_i(B)]$$
$$K_{max_i} = \pi_1 \circ s_{2_\triangleright} \circ [\pi_i(A) \circledast \pi_i(B)]$$

$$K_{BlendedC1} = \circledast_{i=1}^n [\mathbf{1}_{+_i} \circ [[K_{min_i} \circledast K_{max_i}] \circ U[0,1]]]. \quad (22)$$

**Proof.** $K_{BlendedC1}$ is defined in terms of projection kernel, join-kernels, kernel composition, arithmetic kernel and sorts kernel. $\square$

---

**Algorithm 7** Blended Crossover-BLENDEDC1

---

BLENDEDC1$_\alpha(A, B)$

1: $Q = \{\}$
2: **for** $i = 1$ to $n$ **do**
3:      $x_{min} = min(\pi_i(A), \pi_i(B))$
4:      $x_{max} = max(\pi_i(A), \pi_i(B))$
5:      $dx = x_{max} - x_{min}$
6:      $\beta \sim [0, 1]$
7:      $Q_i = \beta * (x_{min} - \alpha * dx) + (1 - \beta) * (x_{max} + \alpha * dx)$
8: **return** $(Q)$

---

7.      A **Linear Crossover** scheme, (LINEARC1: $\Omega^n \times \Omega^n \to \Omega^n \times \Omega^n \times \Omega^n$). The definitions can be seen in Algorithm 8. These methods can be characterized by the kernels $K_{LinearC1}: (\Omega^n \times \Omega^n) \times (\Sigma^{\otimes n} \times \Sigma^{\otimes n} \times \Sigma^{\otimes n}) \to [1, 0]$, defined by:

$$K_{LinearC1} = \circledast_{i=1}^{n}[[\mathbf{1}_{+1_i} \circ [\pi_i(A) \circledast \pi_i(B)]] \circledast \mathbf{1}_{+2_i} \circ [\pi_i(A) \circledast \pi_i(B)] \circledast \mathbf{1}_{+3_i} \circ [\pi_i(A) \circledast \pi_i(B)]]. \tag{23}$$

**Proof.**    $K_{LinearC1}$ is defined in terms of kernel, join-kernels, kernel composition and addition kernel.    $\square$

---

**Algorithm 8** Linear Crossover-LINEARC1

---

LINEARC1$(A, B)$

1: $Q_1 = \{\}$
2: $Q_2 = \{\}$
3: $Q_3 = \{\}$
4: **for** $i = 1$ to $n$ **do**
5:      $Q_{1_i} = (1/2) * \pi_i(A) + (1/2) * \pi_i(B)$
6:      $Q_{2_i} = (3/2) * \pi_i(A) - (1/2) * \pi_i(B)$
7:      $Q_{3_i} = (-1/2) * \pi_i(A) + (3/2) * \pi_i(B)$
8: **return** $(Q_1, Q_2, Q_3)$

---

*4.3. Simulated Annealing (*SA*)*

4.3.1. Concept

The Simulated Annealing algorithm (SA) considers the idea behind the process of heating and cooling a material to recrystallize it, see Algorithm 9. When the temperature decreases, the material settles into a more ordered state, and the state into which they settle is not always the same. This state tends to have low energy compared when the material is in the presence of high temperature ([25]). If we consider energy as a cost function, we can use this approach to minimize cost functions. Therefore, SA is a stochastic algorithm that works with a single-individual that generates a single candidate-solution $x$ (parent) and sets a high temperature to explore the search space. Then, a variation mechanism generates a new candidate-solution $y$ (child) and measures its cost. A replacement policy, that fitness function and the temperature, picks one individual between the father and the child. Finally, a process decreases the temperature looking for each new solution having less energy.

Clearly, the replacement policy in Algorithm 9 (lines 6, ..., 11) is not elitist. This allows SA to expand the search but can lead to the loss of some good candidate-solutions. In practice, it is normal to keep track of the best solution found so far [25]. If this is done, the replacement policy is an elitist one.

---

**Algorithm 9** Simulated Annealing [25]

---

SIMULATED ANNEALING

　1: $T =$ initial temperature $> 0$
　2: $\alpha(T) =$ cooling function: $\alpha(T) \in [0, T]$ for all $T$
　3: Initialize a candidate solution $x_0$ to minimization problem $f(x)$
　4: **while** ¬TERMINATIONCONDITION() **do**
　5:　　Generate a candidate solution $x$
　6:　　**if** $f(x) < f(x_0)$
　7:　　　　$x_0 = x$
　8:　　**else**
　9:　　　　$r = U[0, 1]$
　10:　　　　**if** $r < exp[(f(x_0) - f(x))/T]$
　11:　　　　　$x_0 = x$
　12:　　$T = \alpha(T)$

---

### 4.3.2. Formalization

To formalize and characterize (SA), we use the approach proposed by [11]. We rewrite Algorithm 9 in terms of individual non-stationary stochastic methods, see Algorithm 10. This new Algorithm is in terms of VARIATION-REPLACEMENT methods. Observe that Algorithms 9 and 10 are equivalents. Line 5 of Algorithm 9 is the method VARIATE$_{SA}$ (line 1) of Algorithm 10; lines 6 to 11 of Algorithm 9 is the method REPLACE$_{SA}$ (line 2) of Algorithm 10. Finally, line 12 of Algorithm 9 and method UPDATEPARAMETERS (line 3) perform the same task.

---

**Algorithm 10** Simulated Annealing in terms of **VR** methods

---

NEXTPOP$_{SA}$(x)

　1: $y =$VARIATE$_{SA}$(x)
　2: $y =$REPLACE$_{SA_T}$(y,x)
　3: UPDATEPARAMETERS(T)
　4: **return** $x'$

---

Now, we focus on characterizing (SA) as a VR stochastic method and analyzing its convergence through non-stationary Markov kernels.

**Proposition 6.** *If* REPLACE$_{SA}(x, x)$ *is an elitist method, then it can be characterized by the Markov Kernel* $R_{SA} : \Omega^2 \times \Sigma \longrightarrow [1, 0]$ *defined as:*

$$K_{R_{SA}} = \pi_1 \circ s_2. \tag{24}$$

**Proof.** $K_{R_{SA}}$ is defined in the same way that the method of R$_{HC}$ in [11]. So the proof uses the same argument that Lemma 75 in [11]. □

**Proposition 7.** *If the stochastic method* VARIATE$_{AS_T}$ *can be characterized by a non-stationary Markov kernel* $V_{SA_T}^{(t)} : \Omega \times \Sigma \longrightarrow [1, 0]$ *and condition of Proposition 6 are fulfilled then method the* NEXTPOP$_S A(x)$ *can be described as a* **VR** *non-stationary Markov Kernel defined as*

$$K_{SA}^{(t)} = K_R \circ K_{V_{SA_T}}^{(t)}. \tag{25}$$

**Proof.** $K_{SA}^{(t)}$ is a kernel composition under the given conditions. □

**Proposition 8.** *If* REPLACE$_{SA}$ *is an elitist method, then* NEXTPOP$_{SA}$ *can be characterized by an elitist non-stationary Markov kernel.*

**Proof.** This proof uses the same argument as Proposition 77 in [11]. □

4.3.3. Convergence

**Corollary 3.** *If the conditions of Propositions [6], [7] and [8], are fulfilled and method* VARIATE$_{AS_T}$ *is optimal strictly bounded from zero then* NEXTPOP$_{SA}$ *is optimal strictly bounded from zero.*

**Proof.** Follows from Definition 67, Lemma 68, and Definition 69 in [11] and Proposition [8] that state that NEXTPOP$_{SA}$ can be characterized by an elitist kernel, and this is optimal strictly bounded from zero. □

**Theorem 7.** SA *will converge to the global optimum if* REPLACE$_S A$ *is elitist and if* VARIATE$_{AS_T}$ *is optimal strictly bounded from zero.*

**Proof.** Follows from Corollary [3], and Propositions [6]–[8]. □

*4.4. Evolutionary Strategies (*ES*)*

4.4.1. Concept

Evolutionary Strategies $(\mu/\rho + \lambda)$-ES are a type of Evolutionary Algorithms that apply mutation, recombination, and selection operators to a population of individuals [22], see Algorithm [11]. Every individual has two parts: the candidate solution ($x$) and the set of endogenous strategy parameters ($s$) used to control the mutation operator ([22]). An ES randomly initializes the population, (Line 2), and evolves both parts of the individual (Lines 5-9) up to certain ending-condition is fulfilled (Line 3). The set of endogenous parameters are exposed to evolution (Lines 6 and 8) before producing a child candidate solution (Line 7 and 9) to introduce variety. The new individual is a composition of a set of selected candidate solutions (Line 5). ES generates a new population of $\lambda$ new individuals each generation (Line 4). Finally, ES selects a final population using two possible approaches. The $(\mu + \lambda)$-ES approach that selects the best $\mu$ individuals among the $\mu$ parents and $\lambda$ children or the $(\mu,\lambda)$-ES that selects the best $\mu$ individuals from the $\lambda$ children (notice that $\lambda \geq \mu$ in this case). In this work, we study both of them.

---

**Algorithm 11** Evolutionary strategies described by [22]

---

ES$\mu/\rho \overset{+}{,} \lambda$

1: $g = 0$
2: INITIALIZE($P_q^{(0)} := \{(y_m^{(0)}, s_m^{(0)}, F(y_m^{(0)})), m = 1, \ldots, \mu\}$)
3: **while** ¬TERMINATIONCONDITION() **do**
4:     **for** $l = 1$ to $\lambda$ **do**
5:         $a_l =$ MARRIAGE($P_q^g, \rho$)
6:         $s_l =$ RECOMBINATION$_s(a_l)$
7:         $y_l =$ RECOMBINATION$_y(a_l)$
8:         $s_l' =$ MUTATION$_s(s_l)$
9:         $y_l' =$ MUTATION$_s(y_l, s_l')$
10:         $F_l' = F(y_l')$
11:     $P_0^g = \{(y_l', s_l', F_l'), l = 1, \ldots, \lambda\}$
12:     **if** $(\mu, \lambda)$ **then**
13:         $P_q^{g+1} =$ SELECTION($P_0^g, \mu$)
14:     **else** $(\mu + \lambda)$
15:         $P_q^{g+1} =$ SELECTION($P_0^g, P_q^g, \mu$)
16:     $g = g+1$

---

### 4.4.2. Formalization

To formalize and characterize $(\mu/\rho \overset{+}{,} \lambda)$-ES, we rewrite Algorithm 11 in terms of individual non-stationary stochastic methods, see Algorithm 12. This follows the approach in [11] that express the algorithms in terms of Variation-Replacement methods to study their convergence properties.

Notice that Algorithms 11 and 12 are equivalents: Lines 4–11 in Algorithm 11 is method VARIATE(P) (Line 1) in the NEXTPOP method of Algorithm 12. Also, Lines 12–15 in Algorithm 11 are Line 2 in the NEXTPOP method of Algorithm 12. Using this characterization, we proceed to characterize each method of Algorithm 12 through non-stationary Markov kernels.

With the object of characterizing $(\mu/\rho \overset{+}{,} \lambda)$-ES we need to establish some non-stationary Markov kernels. First, we study the VARIATE method (Line 1, method NEXTPOP, Algorithm 12).

Following Definition 55 in [11], we can express the variation method VARIATE: $\Omega^\mu \longrightarrow \Omega^\lambda$ as a joined stochastic method.

$$\text{VARIATE}(P) = \prod_{i=1}^{\lambda} \text{NEXTSUBPOP}_i(P) \tag{26}$$

where NEXTSUBPOP: $\Omega^\mu \longrightarrow \Omega$ chooses $\rho$ individuals from the population, combines the $\rho$ individuals, generates a child and finally mutates the strategy and the child.

---

**Algorithm 12** Evolutionary strategies algorithm-NextPop method described in terms of VR methods

---

NEXTSUBPOP$_i$(P)

1: $a = $ PICKPARENTS$(P)$
2: $q = $ XOVER$_a(P)$
3: UPDATESTRATEGIES$_a(\text{s}, i)$
4: $d = $ VARIATE$_s(q)$
5: **return** $d$

UPDATESTRATEGIES$_a(s, i)$

1: $z = $ XOVERSTRATEGIE$_a(s)$
2: $s_i = $ VARIATESTRATEGIE$(z)$

VARIATE(P)

1: **for** $i = 1$ to $\lambda$ **do**
2:     $Q_i = $ NEXTSUBPOP$_i(P)$
3: **return** Q

NEXTPOP$_\Psi$(P)

1: $Y = $ VARIATE$(P)$
2: $Q = $ REPLACE$_\Psi(P, Y)$
3: **return** $Q$

---

**Proposition 9.** *If Lines 8 and 9 of method* UPDATESTRATEGIES *of Algorithm 12 can be characterized by non-stationary kernels* $X: \mathbb{R}^\rho \times \mathscr{B}(\mathbb{R})^{\otimes \rho} \longrightarrow [0,1]$ *and* $VS^{(t)}: \mathbb{R} \times \mathscr{B}(\mathbb{R}) \longrightarrow [0,1]$ *respectively.* UPDATESTRATEGIES *can be characterized by a non-stationary kernel* $US^{(t)}: \mathbb{R}^\rho \times \mathscr{B}(\mathbb{R}) \longrightarrow [0,1]$ *defined as:*

$$K_{US}^{(t)} = K_{VS}^{(t)} \circ K_{XS}. \tag{27}$$

**Proof.** $K_{US}^{(t)}$ is in terms of kernel composition, follows from Definition 25 in [11]. $\quad \square$

**Proposition 10.** *If Lines 2 and 4 of Algorithm 12 can be characterized by non-stationary Markov kernels* XOVER$_a$: $(\Omega^\rho \times \Sigma) \longrightarrow [0,1]$ *and* VARIATE$_s$: $(\Omega \times \Sigma) \longrightarrow [0,1]$ *respectively, then the method* NEXTSUBPOP *can be characterized by the kernel* NEXTSUBPOP: $(\Omega^\rho \times \Sigma) \longrightarrow [0,1]$ *defined as the non-stationary kernel:*

$$K_{\text{NEXTSUBPOP}} = K_{\text{VARIATE}_s}^{(t)} \circ K_{XOVER} \circ \pi_{\{1,\dots,\rho\}} \circ K_{\mathcal{P}}. \tag{28}$$

**Proof.** $K_{\text{NEXTSUBPOP}}$ is in terms of kernel composition, follows from Definition 25 in [11]. $\square$

**Proposition 11.** *If* NEXTSUBPOP *can be characterized by a non-stationary Markov kernel, the stochastic method* VARIATE$^{(t)}$ *can be characterized by a kernel* $V$: $\Omega^\mu \times \Sigma^{\otimes \mu} \longrightarrow [0,1]$ *defined as*

$$K_{\text{VARIATE}}^{(t)} = [\otimes_{i=1}^{\lambda}[K_{\text{NEXTSUBPOP}_i}]]. \tag{29}$$

**Proof.** $K_{\text{VARIATE}}^{(t)}$ is a join stochastic method, follows from Definition 55 and Proposition 56 in [11]. $\square$

**Proposition 12.** *The stochastic method* REPLACE$_{(\mu+\lambda)}$ *used in Line 2 of method* NEXTPOP*, can be characterized by the kernel* $R_{\mu,\mu+\lambda}$: $\Omega^{\mu+\lambda} \times \Sigma^{\otimes \mu} \longrightarrow [0,1]$ *defined as* $K_{R_{\mu,\mu+\lambda}} = \pi_{\{1,\dots,\mu\}} \circ s_{\mu+\lambda,\mu+\lambda-1}$ *and the stochastic method* REPLACE$_{(\mu,\lambda)}$*, can be characterized by the kernel* $R_{\mu,\lambda}$: $\Omega^\lambda \times \Sigma^{\otimes \mu} \longrightarrow [0,1$ *defined as* $K_{R_{\mu,\lambda}} = \pi_{\{1,\dots,\mu\}} \circ s_{\lambda,\lambda-1}$ .

**Proof.** $K_{R_{\mu,\lambda}}$ and $K_{R_{\mu+\lambda}}$ are kernels composition. Follows from Definition 25 in [11]. $\square$

**Corollary 4.** *If methods* PICKPARENTS*,* XOVER$_a$*,* XOVERSTRATEGIE$_a$*,* VARIATESTRATEGIE *and,* VARIATE$_s$ *can be described by Markov kernels fulfilling the conditions of Propositions 9 and 10, evolutionary Strategies can be described by a* **VR** *kernel.*

$$K_{ES} = K_R \circ K_V$$

*where:*

$$K_V = K_{\text{VARIATE}}$$

$$K_R = K_{R_{\mu,\lambda}} \text{ or } K_R = K_{R_{\mu+\lambda}}. \tag{30}$$

**Proof.** Follows from Propositions 9–12. $\square$

### 4.4.3. Convergence

**Proposition 13.** *The* NEXTPOP$_{(\mu/\rho+\lambda)-ES}$ *is an elitist stochastic method that can be characterized by an elitist stochastic kernel.*

**Proof.** Let $k \in [1,\mu]$ be the index of the best individual in population $P$, then $f(BEST(P)) = f(P_k)$. Since $P \subseteq \{P \cup \text{VARIATE}(P)\}$ and the method REPLACE is elitist. It is clear that $f(BEST(P \cup \text{VARIATE}(P))) \trianglelefteq f(P_k)$. $\square$

**Corollary 5.** *If conditions of Proposition 9 and 10 are satisfied and* VARIATE$_s$ *is optimal strictly bounded from zero then the method* NEXTPOP$_{\mu+\lambda}$ *is optimal strictly from zero.*

**Proof.** Follows from Definition 67, Lemma 68, and Definition 69 of [11] and Proposition 13 that establish that an elitist kernel is optimal strictly bounded from zero. $\square$

**Theorem 8.** *($\mu/\rho + \lambda$)-ES will converge to the global optimum if methods* PICKPARENTS *and* VARIATE$_s^{(t)}$ *can be characterized by stationary or non-stationary Markov kernels and* VARIATE$_s$ *is optimal strictly bounded from zero.*

**Proof.** Follows from Theorem 3 and Corollary 5. □

## 5. Discussion

We have generalized the conditions of convergence to the global optimum from stationary to non-stationary Markov process that are presented in the work of stochastic global optimization algorithms: a systematic approach proposed in [11]. We study the necessary theory to describe with kernels some arithmetic methods. For doing so, it was necessary to use the concepts studied in real analysis, such as arithmetic between measurable functions. However, the literature found only studied the case of operating two functions on the same variable but not on two different variables. Hence, the concepts of product sigma algebra presented by Gomez in [11], were used to prove that arithmetic operations between measurable functions on two different variables are also measurable.

We formalized some selection and recombination schemes to generalize the theory to cover as many variations of each algorithm as possible. For that, we have found that most of these methods could be characterized using the kernels studied in [11] and the new kernels studied in this paper. This makes us think that other schemes in the literature could be easily adapted to the concepts developed in this paper.

In this paper, we have formalized and characterized the simulated annealing algorithm and evolutionary strategies using the developed theory (both have been formalized in terms of Variation-Replacement kernels). A wide variety of non-stationary algorithms described algorithmically can be found in the literature. However, the theory described in this work cannot be used directly. For that reason, the first step to take is to write the algorithms in terms of Variation-Replacement as shown in Sections 4.3 and 4.4. This approach can also be studied in [1]. There, the class of hybrid adaptive evolutionary algorithms is characterized.

Also, we formulated a set of conditions that SA and ES algorithms should fulfill to achieve a global convergence. After characterizing these algorithms by a Variation-Replacement Kernel, it has been proven that these can converge to the global optimum if the particular implementation of the Variational method is strictly bounded from zero, which depends of the way each algorithm is implemented.

Our future work will focus on using the proposed approach to formalize as many stationary and non-stationary SGOALsas possible, and extending and developing the theory for several particular methods (Mutation, recombination and selection) that can be considered in SGOALs. Moreover, we will study new convergence conditions and not only for the global optimum.

**Author Contributions:** Conceptualization, J.G. and A.R.; methodology, J.G. and A.R.; validation, J.G. and A.R.; formal analysis, J.G. and A.R.; investigation, J.G. and A.R.; writing—original draft preparation, J.G. and A.R.; writing—review and editing, J.G. and A.R.; supervision, Gomez; project administration, Gomez.; All authors have read and agreed to the published version of the manuscript.

**Funding:** This research was funded by Universidad Nacional de Colombia.

**Institutional Review Board Statement:** Not applicable.

**Informed Consent Statement:** Not applicable.

**Data Availability Statement:** Not applicable.

**Conflicts of Interest:** The authors declare no conflict of interest.

## Abbreviations

The following abbreviations are used in this manuscript:

SGOAL     Stochastic Global Optimization Algorithm
SGOALs   Stochastic Global Optimization Algorithms
ES         Evolutionary Strategies
SA        Simulated Annealing

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
