# Peer review of "Non-Stationary Stochastic Global Optimization Algorithms"

_algorithms, doi:10.3390/a15100362_

Round 1

Reviewer 1 Report

The Article “Non-Stationary Stochastic Global Optimization Algorithms” by Jonatan Gomez and Andres Rivera is a theory paper based on the authors’ previous works. In the abstract, they declare their goal of this article as to firstly “generalize the sufficient conditions convergence from stationary to non-stationary Markov processes”, secondly to “develop the necessary theory to define kernels for arithmetic operations between measurable functions”, and thirdly, to “develop Markov kernels for some selection and recombination schemes“, which is a fitting summary of this paper.

In the abstract, the authors declare this goal and introduce the works on which this work is based. However, the first word of this article is the name of one of the authors, as the paragraph starts with a reference to the authors' previous works. The paragraph should be reformulated to gain focus on the topic immediately.

Section 0 ‘Introduction’ starts with a brief introduction to the field of interest and also motivates this work. Previous and related work that does not originate from the authors themselves is not presented, nor is there any differentiation of this work from other work. It remains unclear whether there are other works in this field and what is state of the art. 

After this short introduction, a review of the concepts proposed in an earlier paper is given. This part is a pure repetition of content published in an earlier paper and should, if repeated here, move to a new section, ‘Preliminaries’.

Some trivialities that stood out while reading:

  • The abbreviation SGOALs is used in lines 19 and 24 but introduced only in line 36.

  • The term NEXTPOP is used in line 38 and explained in line 58 only

  • Section 0.1 Starts with ‘here’ better use: in this subsection or similar

  • The sentence in lines 65/66 needs to be rephrased.

  • Line 16 help+s+

  • Line 21 prove+s+

In Section 1, ‘Materials and Methods’, and Section 2, ‘Results’, the main work of this paper is introduced.

  • In line 259, the expression “iff” is used; here, the wording "one and only one" should perhaps be used preferably.

  • In line 340/341, please add further references

  • In line 308/309, the sentence is missing an article and at least one s: generate+s+ (..?..) offspring(+s+?)

Section 2

Section 3 ‘Discussion’ lacks any discussion of the work presented here. In this short paragraph, only the presented work is summarized, and finally, an outlook on possible future work is provided. 

  • Line 539: that are present+ed+ in the work

References

  • typo in [1] line 569

  • bad linebreak in line 582

Author Response

Thank you very much for your comments.

Point 1: In the abstract, the authors declare this goal and introduce the works on which this work is based. However, the first word of this article is the name of one of the authors, as the paragraph starts with a reference to the authors' previous works. The paragraph should be reformulated to gain focus on the topic immediately.

 We have reformulated the abstract. 

Point 2: Section 0 ‘Introduction’ starts with a brief introduction to the field of interest and also motivates this work. Previous and related work that does not originate from the authors themselves is not presented, nor is there any differentiation of this work from other work. It remains unclear whether there are other works in this field and what is state of the art. 

We have reformulated the introduction according to your suggestions, We have added some related work and what is new in our work. Please see the introduction. 

Point 3: After this short introduction, a review of the concepts proposed in an earlier paper is given. This part is a pure repetition of content published in an earlier paper and should, if repeated here, move to a new section, ‘Preliminaries’.

We have created a preliminaries section. 

Point 4: Some trivialities that stood out while reading:

  • The abbreviation SGOALs is used in lines 19 and 24 but introduced only in line 36.  It is already corrected. It is defined in both: the abstract and in the introduction.

  • The term NEXTPOP is used in line 38 and explained in line 58 only. We have changed the term NEXTPOP to next population in this section

  • Section 0.1 Starts with ‘here’ better use: in this subsection or similar. We have changed  this section according to your comments. 

  • The sentence in lines 65/66 needs to be rephrased. We have changed this  sentence. 

  • Line 16 help+s+ Not applicable, because we have changed the introduction. 

  • Line 21 prove+s+ Not applicable, because we have changed the introduction. 

Point 5: In Section 1, ‘Materials and Methods’, and Section 2, ‘Results’, the main work of this paper is introduced.

  • In line 259, the expression “iff” is used; here, the wording "one and only one" should perhaps be used preferably. We have changed "iff" to "if and only if" because if one condition is true the other ones are also true. So, we think "one and only one" in this case is not adequate.

  • In line 340/341, please add further references. We have added more references in this section. 

  • In line 308/309, the sentence is missing an article and at least one s: generate+s+ (..?..) offspring(+s+?). We have changed that sentence. 

Point 5: Section 2

Point 6: Section 3 ‘Discussion’ lacks any discussion of the work presented here. In this short paragraph, only the presented work is summarized, and finally, an outlook on possible future work is provided. 

We have changed the discussion. Please,  see the new version.

  • Line 539: that are present+ed+ in the work, This was corrected.

Point 7:  References

  • typo in [1] line 569 We could not find the typo. Maybe the capital letter after the colon. However, this is the exact name of the paper.

  • bad linebreak in line 582 Fixed

Reviewer 2 Report

The research describes the viewpoint of studying SGO algorithms form a Markovian perspective. As such, this is an interesting question, which this reviewer has seen before in literature on SGO. However, it seems that the author created his own terminology and only goes deeper in his own thoughts, without looking around what has been going on in literature on Stochastic Global optimization for the last 60 years. To share the ingenious thoughts, it would be good to have a look at the terminology and theory developed before. 

 Please, do not start a paper with a reference. Please, do not start with a reference to yourself and if there are references, embed your question in the literature on Stochastic Global Optimization. For instance the papers published on this topic by Marco Locatelli, Eligius Hendrix and the books of Anatoly Zhigliavski and Antanas Zilinskas and the many references therein. You will become aware, that the posed questions have been of interest already in last century. After the introduction in Section 0., please tell the reader how the rest of the paper is organized and start a new section. Avoid sub-sub-sub-sectioning.

 In (1), please provide the ideas you have about the feasible set. Is it a finite set, a set with a non-empty interior or simply a box constrained area? Notice that (1) is not conventional. It is not clear one aims at finding the set of global minimum points, or one of the global minimum points. Suggest to use \min or \argmin.

The way you describe Best(P_t), implies you are only looking for one global minimum point approximation and do not aim the population to converge the set of global minimum point how is done in Controlled Random Search and Uniform Covering by Probabilistic Rejection.

 There is some confusion in mathematical notation, but also in sentences that include equations.

 We insist on reading literature on Stochastic Global Optimization, rather than referring to a 21st century 4 page proceedings contribution. Lemma 2 is the concept called by Törn and Zilinskas as everywhere dense sampling. As long as each point in the feasible space has a positive probability density to be reached, an SGO algorithm will converge in function value to the global minimum point. So please, consider Törn and Zilinskas 1989, Zhigliavsky and Zilinskas 2008 and/or Hendrix and Toth 2010.

 With respect to the analysis of simulated annealing as a Markovian process, we refer to the work of Edwin Romeijn about 30 years ago.

Author Response

Thank you very much for your suggestions and comments. They have been of great value to improve the paper.  We think that all the comments you made about the paper have been taken into account (Including the corrections made on the PDF). Also, we have changed the discussion. Please, check the new version.

P1: The research describes the viewpoint of studying SGO algorithms form a Markovian perspective. As such, this is an interesting question, which this reviewer has seen before in literature on SGO. However, it seems that the author created his own terminology and only goes deeper in his own thoughts, without looking around what has been going on in literature on Stochastic Global optimization for the last 60 years. To share the ingenious thoughts, it would be good to have a look at the terminology and theory developed before. 

We have reformulated the introduction according to your suggestions. We have added related work and what is new in our work. Please see the introduction. 

P2: Please, do not start a paper with a reference. Please, do not start with a reference to yourself and if there are references, embed your question in the literature on Stochastic Global Optimization. For instance the papers published on this topic by Marco Locatelli, Eligius Hendrix and the books of Anatoly Zhigliavski and Antanas Zilinskas and the many references therein. You will become aware, that the posed questions have been of interest already in last century. After the introduction in Section 0., please tell the reader how the rest of the paper is organized and start a new section. Avoid sub-sub-sub-sectioning.

We have changed our abstract to delete the reference according to your suggestion. As we said before, we have added related work and what is new in our work. Please, see the introduction. We have changed section 0 to section 1. At the end of the introduction, we describe how the paper is organized. Regarding the sub-sub-section suggestions, we used this format so that it would fit the sections required by the journal (introduction, methods, results, and discussion). What we have done is to move the previous work to a new section called preliminaries.

P3: In (1), please provide the ideas you have about the feasible set. Is it a finite set, a set with a non-empty interior or simply a box constrained area? Notice that (1) is not conventional. It is not clear one aims at finding the set of global minimum points, or one of the global minimum points. Suggest to use \min or \argmin.

We have changed this equation according to your suggestion. Please see the new version of the paper. The feasible set we would like to let be as generic as possible.

P4:The way you describe Best(P_t), implies you are only looking for one global minimum point approximation and do not aim the population to converge the set of global minimum point how is done in Controlled Random Search and Uniform Covering by Probabilistic Rejection.

We have changed the description of the BEST method.

P5: There is some confusion in mathematical notation, but also in sentences that include equations. 

We have changed the sentences and notation you highlighted in the PDF.

P6: We insist on reading literature on Stochastic Global Optimization, rather than referring to a 21stcentury 4 page proceedings contribution. Lemma 2 is the concept called by Törn and Zilinskas as everywhere dense sampling. As long as each point in the feasible space has a positive probability density to be reached, an SGO algorithm will converge in function value to the global minimum point. So please, consider Törn and Zilinskas 1989, Zhigliavsky and Zilinskas 2008 and/or Hendrix and Toth 2010.

In the introduction we have consider several of these references. Also, we have changed Lemma 2 according to your suggestion.

P7: With respect to the analysis of simulated annealing as a Markovian process, we refer to the work of Edwin Romeijn about 30 years ago.

There are several studies in the literature about simulated annealing. However, we chose this algorithm to give an example of how to use the concepts developed in this paper.

Round 2

Reviewer 1 Report

Point 7: References

  • typo in [1] line 569 We could not find the typo. Maybe the capital letter after the colon. However, this is the exact name of the paper.

This comment was referring to the DOI Link. While the link is working fine, the link text says "https://doi.org/https://doi.org/10.1016/j.ins.2018.09.021"

Author Response

  • typo in [1] line 569 We could not find the typo. Maybe the capital letter after the colon. However, this is the exact name of the paper.

This comment was referring to the DOI Link. While the link is working fine, the link text says "https://doi.org/https://doi.org/10.1016/j.ins.2018.09.021"

Fixed.

Thank you very much for all your suggestions.

Reviewer 2 Report

The manuscript has been improved considerably and the message is coming out much better. I provide a list of comments on the (SP)English mainly.

Author Response

The manuscript has been improved considerably and the message is coming out much better. I provide a list of comments on the (SP)English mainly.

We want to thank you again for your suggestions and comments. They have been invaluable in improving the paper.

The suggestions written in the PDF were taken into account. Please, check the new version.

Round 3

Reviewer 1 Report

The manuscript has been improved considerably

Reviewer 2 Report

Good modification of the story. Hope the rest of the world is as enthusiastic as the authors about it.